# Apoptotic Bodies: Particular Extracellular Vesicles Involved in Intercellular Communication

**DOI:** 10.3390/biology9010021

**Published:** 2020-01-20

**Authors:** Michela Battistelli, Elisabetta Falcieri

**Affiliations:** Department of Biomolecular Sciences (DiSB), Urbino University Carlo Bo, 61029 Urbino (PU), Italy; elisabetta.falcieri@uniurb.it

**Keywords:** apoptotic bodies, extracellular vesicles, ultrastructure, intercellular communication

## Abstract

In the last decade, a new method of cell–cell communication mediated by membranous extracellular vesicles (EVs) has emerged. EVs, including exosomes, microvesicles, and apoptotic bodies (ApoBDs), represent a new and important topic, because they are a means of communication between cells and they can also be involved in removing cellular contents. EVs are characterized by differences in size, origin, and content and different types have different functions. They appear as membranous sacs released by a variety of cells, in different physiological and patho-physiological conditions. Intringuingly, exosomes and microvesicles are a potent source of genetic information carriers between different cell types both within a species and even across a species barrier. New, and therefore still relatively poorly known vesicles are apoptotic bodies, on which numerous in-depth studies are needed in order to understand their role and possible function. In this review we would like to analyze their morpho-functional characteristics.

## 1. Introduction

### 1.1. Extracellular Vesicles

The direct contact between two or more cells and the interaction between the ligand and the receptor have been considered, for a long time, the predominant means for intercellular communication [1]. Recently, a new possibility of cell–cell communication mediated by membranous extracellular vesicles (EVs) has been proposed and is the object of a number of molecular and morpho-functional studies. Extracellular vesicles are signaling-correlated organelles, released by many cell types and highly conserved in both prokaryotes and eukaryotes. All organisms produce extracellular vesicles and both normal and dying cells release membrane-bound vesicles such as exosomes, microvesicles, and apoptotic bodies [2,3,4]. Recent studies have demonstrated that extracellular vesicles can cause both protective and pathologic effects, depending on the precise condition. They can be detected in body fluids including urine, saliva, blood, plasma, amniotic fluid, breast milk, pleural ascites, synovial fluid, and cerebral spinal fluid [5,6,7].

Extracellular vesicles can encapsulate small portions of the subjacent cytosol, creating a heterogeneous population of phospholipid-walled vesicles, which play an important role in intercellular communication, immune response, angiogenesis, and signal transduction, in both physiological and pathological conditions [8].

Extracellular vesicles are also referred to as microparticles, microvesicles, microsomes, lipid vesicles, apoptotic blebs, or exosomes, depending on the basis of their biogenesis or release pathways [9,10,11].

EVs are characterized by their size (40–100 nm for exosomes, 100–500 nm diameter for the larger microvesicles, and 500 nm–2 µm for apoptotic bodies), by their cells of origin, such as megakaryocytes, platelets, red blood cells, and endothelial cells [12,13], and by their intravesicular contents. Their inner content depends on their cells of origin and can include proteins, sugars, lipids, adhesion integrins, growth factors, protease inhibitors, and different types of genetic material such as double stranded DNA, mRNA, or microRNA [14] (Figure 1).

### 1.2. Brief Presentation of the Aim of the Review

The purpose of this mini review is to introduce the importance of apoptotic bodies within the extracellular vesicle family. We want to underline the clear difference of these vesicles from exosomes and microvesicles, in terms of structure, size, and formation, while being also, at the same time, certainly involved in cellular communication.

## 2. Apoptosis

Many different cell types undergo apoptosis, a physiologically-programmed cell death, commonly appearing in multicellular organisms and occasionally also described in yeasts and bacteria [15,16].

Apoptosis is an active process that occurs in normal cell turnover, proper development and functioning of the immune system, hormone-dependent atrophy, embryonic development, and chemical-induced cell death. It plays a crucial role during the process of development and aging and it represents a homeostatic mechanism for maintaining regular cell populations in tissues [17]. Although there is a wide variety of stimuli and conditions, both physiological and pathological, that can trigger apoptosis, not all cells are necessarily sensitive to the same stimuli [18,19].

Apoptosis is characterized by a sequence of steps that lead to the cell deletion. Light and electron microscopy analyses represent an effective approach to evaluate morphological features during the apoptotic process, but only when associated with biochemical study, can a complete understanding of the complex mechanism can be achieved.

Intrinsic pathways or extrinsic stimuli are able to lead to the activation of the apoptotic signaling. Following induction, a caspase-dependent proteolytic cascade is activated. Caspases are aspartic acid-specific proteases responsible for cellular component degradation. Caspase-8 and -9 act as initiators of the apoptotic signaling pathway, while caspases-3, -6, and -7, operate as executor caspases, which actively participate in the degradation of cell substrates.

Caspase activation can follow two main apoptotic pathways, the extrinsic, or death receptor pathway, and the intrinsic, or mitochondrial one. The dying cell is engulfed by professional phagocytes or by neighboring cells. An efficient apoptotic cell removal is driven by interaction with phagocytes through the expression of “eat me” signals and the release of “find me” signals, which facilitate the engulfment of the dying cell and its eventual digestion in their phagolysosomes [20].

Apoptosis progresses through several stages, first nuclear chromatin condensation, then nuclear splitting and the frequent appearance of micronuclei, then membrane blebbing and finally, splitting of the cellular content into distinct membrane-enclosed vesicles, termed apoptotic bodies or, more recently, apoptosome [21,22]. The apoptotic cell disassembly process and the apoptotic material removal by phagocytes are very rapid, therefore the presence of apoptotic bodies (ApoBDs) is very limited in vivo [23], (Figure 2).

Early in apoptotic cell death, there is a very particular nuclear behavior involving the progressive margination and compacting of chromatin, which then characteristically aggregates under the nuclear membrane. This compacting areas of chromatin are rearrangement due to DNA fragmentation, which generally occurs, even if with different behaviors, in dying cells. DNA breaks can be evaluated, in vitro, by means of DNA gel electrophoresis or, in situ, by cleavage points labelling, using TdT or Pol 1, and then analysed by light and electron microscopy [24].

Chromatin margination is first limited to thin electron-dense areas underlying the nuclear envelope. These electron-dens areas are then organized in cap-shaped compact structures [25]. During this phase, an unusual translocation of nuclear pores (np), closed to diffuse chromatin zones, appears, but no pore can be observed around the zone of presence of compact areas. We can evaluate the nuclear pore presence by conventional transmission electron microscopy or by freeze-fracture [26,27,28], (Figure 2).

During chromatin margination plasma membrane blebs appear on the cellular surface [29,30] (Figure 2). This process represents another morphological hallmark of in vitro and in vivo apoptosis and it is controlled by actomyosin contraction [31]. In fact during apoptosis, after caspase activation, cleavage of nuclear lamina A/C and B1 weakens the lamina and makes fragile the nuclear envelope, so allowing its breakdown [32,33].

Even if deep nuclear changes take place, a good, long-lasting cytoplasm and cellular membrane preservation can be observed. Nevertheless, cells undergo characteristic morphological changes in which the cytoskeleton plays an active role. Apoptotic cells are characterized by profound cytoskeleton reorganizations and caspase-mediated digestion of cytoskeleton proteins ensures the proper dismantlement of the dying cell during this process [34,35].

The eukaryotic cytoskeleton is mainly composed of actin filaments, microtubules, and intermediate filaments [3]. These three constituents coordinate to increase tensile strength, allow cell motility, maintain plasma integrity, participate in cell division, contribute to cell morphology, and provide a network for cellular transport. Classically, it has been accepted that microtubules and intermediate filaments are disorganized at the onset of the execution phase, while the actin cytoskeleton is responsible for cell remodeling during it. At later stages, microtubule reorganization gives rise to the apoptotic microtubule network, a structure that sustains apoptotic cell morphology, maintains plasma membrane integrity, and participates in the dispersion of cellular and nuclear fragments [36,37,38,39]. 

The later stages are characterized by the appearance of a variable number of micro-, and macronuclei [40]. This section may be divided by subheadings. It should provide a concise and precise description of the experimental results, their interpretation as well as the experimental conclusions that can be drawn.

## 3. Extracellular Vesicles, Other than Apoptotic Bodies

### 3.1. Exosomes

Exosomes are extracellular vesicles, whose diameter ranges from 40 to 120 nm. They are released upon fusion of multivesicular bodies and the plasma membrane, with final releasing of smaller vesicles [41] (Figure 1). They originate as intraluminal vesicles during the assembling of multivesicular bodies. Exosomes, conserved structures formed in every cell type, were identified over 25 years ago during the differentiation of erythrocytes. Most body fluids contain exosomes. Their contents have been shown to change in various diseases including viral infections, neurodegenerative diseases (prions, Alzheimer, and Huntington disease), and cancer: Hence exosomes are intensively investigated as a source of novel biomarkers [42]. The protein and lipid composition of exosomes reflects their cellular origin. The most common exosomal proteins are annexins, tetraspanins (CD63, CD81, CD82, and CD9), as well as heat-shock proteins (Hsp60, Hsp70, and Hsp90). In 2007, Valadi et al. showed a new mechanism of cell–cell communication i.e. the delivery of RNA by transfer through exosomes [43]. Many studies have confirmed and highlighted that exosomes can carry mRNAs and miRNAs [44]. Exosomes are involved in intercellular communication, to receive and send signals [45,46,47]. 

### 3.2. Microvesicles

Microvesicles bud directly from the plasma membrane, have a 100–500 nm size, and include cytoplasmic material [10] (Figure 1). Their biogenesis occurs via the direct outward blebbing and pinching of the plasma membrane releasing the nascent microvesicle into the extracellular space [48]. They contain various levels of adhesion molecules, such as integrins, that, when released by stem cells at different stages of the cell cycle, could affect the vesicle trafficking and uptake. This is accompanied by distinct, localized changes in protein and lipid components of the plasma membrane, which induce modifications in membrane curvature and rigidity. Microvesicles are membrane vesicles carrying proteins, nucleic acids, and bioactive lipids [49] of the cell of origin. These vesicles, when released within the extracellular space and entered into the circulation may transfer their cargo to neighboring or distant cells, so inducing phenotypical and functional changes, relevant in several physio-pathological conditions [13]. In many biological systems, extracellular vesicles are emerging as important mediators of cell–cell communication cooperating with the maintenance of function [50]. 

## 4. Apoptotic Bodies

Dying cells, release 500 nm–2 μm vesicular apoptotic bodies that can be more abundant than exosomes or MVs under specific conditions and appear variable in size, structure, and composition (Figure 1).

While microvesicles and exosomes can operate as ‘safe containers’ mediating inter-cellular communication, apoptotic bodies appear after the disassembly of an apoptotic cell into subcellular fragments. The formation of ApoBDs is an important process downstream apoptotic cell death, considered a hallmark of apoptosis [51].

Extracellular vesicles are emerging as potent sources of genetic information transfer between mammalian cells and tissues. The less-known apoptotic bodies need numerous and in-depth studies, to understand their role and their possible functions.

As a final phase of apoptotic death, the cell divides into a variable number of apoptotic bodies. These are a peculiar family of extracellular vesicles, quite variable in size and content [4], (Figure 3). 

In fact, they may contain a wide variety of cellular components: micronuclei, chromatin remnants, cytosol portions, degraded proteins, DNA fragments, or even intact organelles.

Many reports describe apoptotic body formation, but little is known about the cellular pathological processes and the morphological changes involved in their function. In particular, it is known that ApoBDs, differently from other microvesicles and contain a large amount of RNA [52]. Moreover, in ApoBDs of larger size, proteins, lipids, RNA, and DNA molecules have been demonstrated [53,54,55]. Presumably, ApoBDs can have important effects on their downstream cells or recipient cells, given their larger molecular pool [25]. Nevertheless, little is known of their function, whereas major scientific progress on the role of EVs in cancer biology has been achieved by investigating exosomes and/or microvesicles [9].

After releasing in extracellular space, ApoBDs are phagocytosed by macrophages, parenchymal cells, or neoplastic cells and degraded within phagolysosomes. Macrophages that engulf and digest apoptotic cells are called “tingible body macrophages”. The tingible bodies are the bits of nuclear debris released from the apoptotic cells [55,56]. 

No inflammatory reaction is associated with apoptosis nor with removal of apoptotic cells because: (1) apoptotic cells do not release free cellular constituents into the surrounding interstitial tissue; (2) ApoBDs are quickly phagocytosed by surrounding cells, thus likely preventing secondary necrosis; and (3) the engulfing cells do not produce anti-inflammatory cytokines [57,58,59]. 

On the other hand, the phagocytosis of apoptotic cells and apoptotic bodies, is precisely coordinated. During the early stages of apoptosis, a membrane lipid rearrangement occurs, which involves phosphatidylserine (PS) translocation from the inner to the outer leaflet [60].

Thus PS, a phospholipid, normally localized in the inner leaflet of the plasma membrane, is remodeled and it is exposed onto the outer leaflet, which is believed to act as an “eat me” signal that facilitates the recognition and uptake of apoptotic cells by phagocytes [61,62]. PS is also externalized under conditions other than apoptosis [63]. In fact, viable cancer cells have been reported to have elevated levels of exposed PS [64] which has been proposed to be involved in modulating cell signaling and vesicle formation [61]. PS exposure has also been observed in some models of tissue differentiation, such as erythroblast maturation [65], where this mechanism has been described to promote nucleus extrusion for the final blood cell formation.

While apoptotic bodies usually expose PS on the outer membrane, several studies have shown exposed PS on microvesicles and exosome surfaces when derived from tumor cells, suggesting further significance as a possible tumoral marker [66].

Apoptotic bodies, “little sealed sacs” containing information and substances from dying cells, were previously regarded as garbage bags until they were discovered to be capable of delivering useful materials to healthy recipient cells (e.g., autoantigens) [23]. The formation of ApoBDs can promote efficient removal of cell debris by means of the surrounding phagocytes. On the other hand, ApoBDs can harbor biomolecules including microRNA [17] and DNA [67] to regulate intercellular communication [20]. In dead cells, ApoBDs, Microvesicles, and Exosome contain fundamentally different RNA profiles; rRNA can be primarily found in Abs [68].

It is now becoming increasingly clear that apoptotic body formation is a result of cell disassembly, a complex process involving highly coordinated morphological steps. Depending on the mechanism used by a particular cell type undergoing apoptotic cell disassembly, a different quantity and quality of ApoBDs will be generated. But it is not yet clear why different cell types need to disassemble differently and the functional significance of such diversity. 

In autoimmune diseases, a defect in the clearance of ApoBDs formation may contribute to the development of autoimmunity [69,70,71]. It has also been demonstrated that ApoBDs seem to have a greater procoagulant effect on cancer cells native cells. These results highlight the potential of ApoBDs to contribute to the prothrombotic state and anticancer immunity [72,73,74].

Even if further studies are mandatory to provide scientific evidence to researchers in biology and medicine, apoptotic body formation represents a process closely involved in both cell clearance and intercellular communication [75,76].

## 5. Conclusions

In this mini review we wanted to focus on apoptotic bodies, which, although commonly mentioned among the micro-vesicles, are little studied. It is really important to consider the entire heterogeneous panel of extracellular vesicles (microvesicles, exosomes, and apoptotic bodies) and to highlight the specific contributions of the different vesicles types to the variety of intercellular communications increasingly involved in human health and diseases. So it will be the goal of our next work to deepen knowledge of the functional morphological characteristics of apoptotic bodies.

## Figures and Tables

**Figure 1 biology-09-00021-f001:**
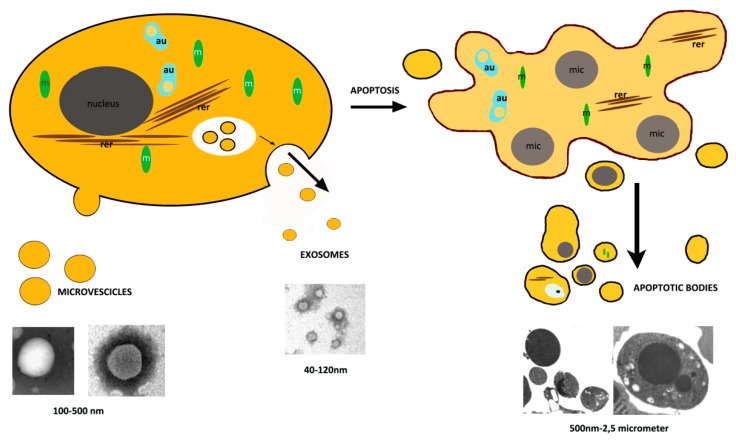
Scheme of extracellular vesicle formation. In this figure shows the biogenesis and release of microvesicles and exosomes. Their morphology was observed by transmission electron microscope (TEM) after negative staining. Apoptotic body extrusion appears in the scheme and in sections of conventionally embedded apoptotic cells. m = mitochondria, rer = rough endoplasmic reticulum, mic = micronuclei, Bar = 200 nm.

**Figure 2 biology-09-00021-f002:**
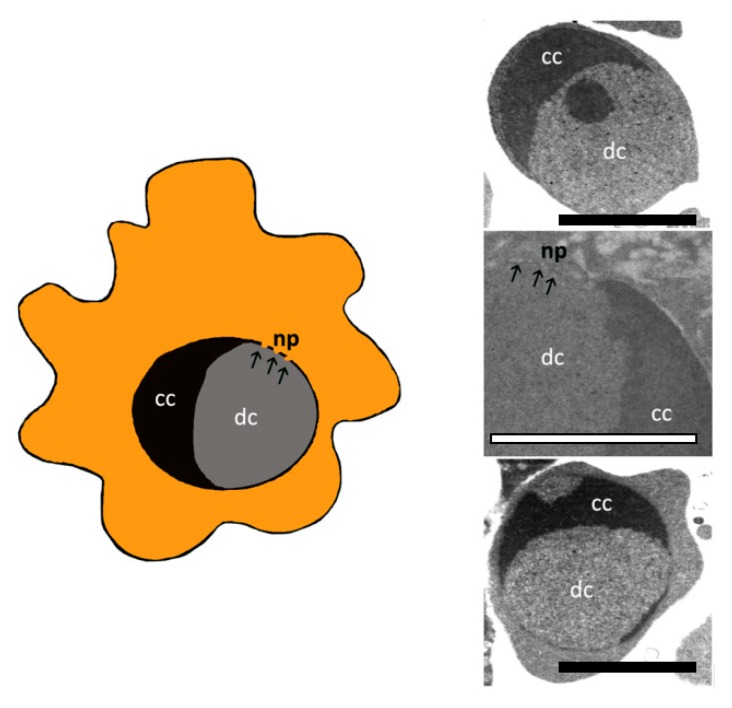
Schematic representation of nuclear changes during early apoptotic phases. Chromatin condensation (cc) and nuclear pore translocation (np) are also seen in TEM observations. dc = diffuse chromatin. Bar = 600 nm.

**Figure 3 biology-09-00021-f003:**
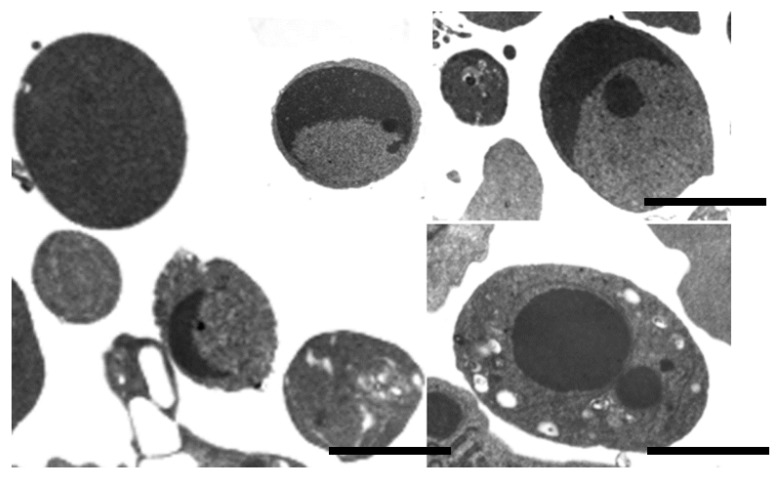
TEM of apoptotic bodies. Bar = 600 nm.

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
