# Peer review of "Apoptotic Bodies: Particular Extracellular Vesicles Involved in Intercellular Communication"

_biology, 2020, doi:10.3390/biology9010021_

Round 1

Reviewer 1 Report

The Article “Apoptotic bodies: particular extracellular vesicles involved in intercellular communication” is a short review about a subtype of the extracellular vesicle family: apoptotic bodies. The authors first briefly described the background of extracellular vesicles and then used three chapters, which are exosome, microvesicles and apoptotic body, to discuss the function and biogenesis of exosomes, microvesicles and apoptotic bodies. The author suggested that the apoptotic body is a kind of vesicle that is deeply involved in morphological steps and pathological processes. Further studies about the apoptotic body formation can represent a process intimately involved in cell clearance and intercellular communication.

1. In the exosome chapter, the author described the size of exosomes is “40 nm to 120 nm,” but in the extracellular vesicles chapter, the author described “30–100nm for exosomes”.Please clarify.
2. Please give more information about the biogenesis of exosomes and microvesicles. This part is a little bit weak.
3. The authors should give more information about what kind of functions or pathways the apoptotic bodies involve. Such as immune cell response regulation [1-3] or promoting the antitumor immunity process [4-6] can be a good topic.

1. Schiller, M., et al., Induction of type I IFN is a physiological immune reaction to apoptotic cell-derived membrane microparticles. J Immunol, 2012. 189(4): p. 1747-56.
2. Berda-Haddad, Y., et al., Sterile inflammation of endothelial cell-derived apoptotic bodies is mediated by interleukin-1alpha. Proc Natl Acad Sci U S A, 2011. 108(51): p. 20684-9.
3. Schiller, M., et al., Autoantigens are translocated into small apoptotic bodies during early stages of apoptosis. Cell Death Differ, 2008. 15(1): p. 183-91.
4. Muhsin-Sharafaldine, M.R., et al., Mechanistic insight into the procoagulant activity of tumor-derived apoptotic vesicles. Biochim Biophys Acta Gen Subj, 2017. 1861(2): p. 286-295.
5. Muhsin-Sharafaldine, M.R., et al., Procoagulant and immunogenic properties of melanoma exosomes, microvesicles and apoptotic vesicles. Oncotarget, 2016. 7(35): p. 56279-56294.
6. Distler, J.H., et al., The release of microparticles by apoptotic cells and their effects on macrophages. Apoptosis, 2005. 10(4): p. 731-41.

Author Response

Response to Reviewer 1

1) We have corrected the measuring range of the exosomes diameter. The correct value is between 40 and 100 micrometers. It was a typing error while writing the paper.

2) We have introduced more information about the biogenesis of exosomes and microvesicles.

3) We have included additional information regarding the structure and function of apoptotic bodies. We also explained the mechanism of their formation. We finally included the inherent references suggested by the Reviewer 1, such as those concerning immune cell response regulation [72-74] and promoting the antitumor immunity process [75-77].

A further english revision has been carried out.

Reviewer 2 Report

The submitted manuscript is quite interesting and well documented, although a few references [see details] may be inappropriately cited.

The aim of entering apoptotic bodies as one of the main components in the Extracellular Vesicles (EV) field with regard to intercellular communication is rather new, so the approach may be qualified as both original and a necessary preliminary attempt to stress the importance of apoptotic bodies into the EV challenging heterogeneity.

However, the actual presentation is not worth the potential interest of the topic : indeed, it is more like a mere listing of sub-topics, without any convincing common thread, and, thus, it gives apparently more importance to EVs - which are already highly described in the current literature - than to apoptosis. Therefore, the paper could be much more attractive and relevant by simply modifying the presentation, in the following way:

Abstract

1. Introduction

With a short historical part highlighting the main discovery hallmarks for “Apoptosis” “Microvesicles (MVs)”, Exosomes, and “Extracellular Vesicles (EVs)”, adding a few pertinent references (see details).

Brief presentation of the aim of the review.

2. Apoptosis

3. Extracellular Vesicles, other than Apoptotic Bodies

 3.1. Microvesicles

       3.2 Exosomes

4. Apoptotic Bodies

5. Conclusion

Stressing the importance of taking into account the whole EV heterogeneous panel (microvesicles, exosomes and apoptotic bodies) to try and solve their respective putative important contributions to the numerous intercellular communications, which are increasingly involved in human health and disease.

Acknowledgments

References

  What do you want to do ? New mailCopy

Author Response

Response to Reviewer 2

1)   We have modified the presentation of the paper, as suggested by the Reviewer 1, to make it more attractive and easy to understand.

2)   We double-checked and expanded the references

3)   We modified the conclusion, in which the potential importance of apoptotic bodies , also as a starting point for future studies, is clearly  emphasyzed.

Round 2

Reviewer 1 Report

All questions/ concerns have been adequately addressed.

Author Response

We have check the english

Reviewer 2 Report

Main comments to Authors:

As anticipated, by a thorough modification of the presentation, the paper is becoming much more attractive. Moreover, the idea of briefly describing the "Extracellular Vesicles" in the Introduction is quite nice, and much better than the previously suggested more historical description. Thus, many of the detailed suggested modifications are no more useful, in order to keep the manuscript with its initial originality.

Minor corrections remain to be performed:

a few writing mistakes in the text and a mistake in Fig. 1 caption: extracellular 

mainly problems with the references  numbers in the new version.

Suggestions for minor corrections in the new version of the manuscript

Manuscript ID: biology-524887
Type of manuscript: Review
Title: Apoptotic bodies: particular extracellular vesicles involved in
intercellular communication.
Authors: Michela Battistelli *, Elisabetta Falcieri
Submitted to section: Cell Biology

----------------------------------------------------------------------------------------------------------

Introduction

- Extracellular vesicles

2 line 3

[2-4] instead of [2, 3, 4]

line 6

[5-7] instead of [5, 6, 7]

line 12

[9-11] instead of [9, 10, 11]

line 18

Ref. [14] is a mix of ref. 14 and ref. 23 (v1), which has been suppressed in the revised version!

Apoptosis

last line

a complete understanding

3 lines 13-14

apoptotic bodies (ApoBDs) or, more recently, apoptosomes [21, 22]. Apoptotic cells disassembly process and apoptotic material removal by phagocytes being very rapid, the presence….

lines 22-23

structures [25]. During this phase an unusual translocation

line 25

fracture [26-28].

line 28

contraction [31]. In fact during apoptosis

4 Extracellular vesicles, other than Apoptotic Bodies

- Exosomes

4 line 15

Ref. [9-11] are probably not appropriate as a reference to exosomes identified over 25 years ago during the differentiation of erythrocytes.

Ref. [9-11] were already cited p. 2 for a definition of different EVs!

line 19

heat-shock proteins

line 23

signals [46-48].

- Microvesicles

line 32

Microvesicles are membrane vesicles carrying, as exosomes, proteins, nucleic acids and bioactive lipids [50] of the cell of origin.

line 36-37

Microvesicles can induce also, cell phenotype modulation[50]. This is also the case for microvesicles.

5 Apoptotic bodies

In v2, ref. 43, 49, 72-77 has been added, but, following the modified presentation, the ref. in v2 match with the ones in v1, up to ref. 71, and 78,79.

However, the citations in the text should be carefully checked, as there are problems with many ref. numbers in this last part of the manuscript:

By comparison with the text in v1, the successive ref. in v2 should be [47] instead of [51], [48] (not 52), [52] (not 53), [53] (not 54) and so on: [54], [55-57], [58], [59,60], [61], [62], [63], [64], [65], [66], [67], [68], 69], (23), [70], [70], (20), [71].

                                   !                                                 ?                 ?       ?      ?

In both [v2] and (v1), the ref. [64] (63) and [67] (66) are the same, but there is a mistake in the name of the first author in ref [64] (63). The right ones are ref. [67] (66)!

line 4

microvesicles and exosomes can operate

lines 8 -9

and tissues resulting. Thus the new and therefore still relatively little The less known vesicles are apoptotic bodies, on which need numerous and in depth studies are neded, to understand their role and their possible therapeutic use.

- Apoptotic bodies     not useful!

line 17

apoptotic bodies, ApoBDs, Abs, or Abs: you have to choose!

lines 18-19

Moreover, in ApoBDs of larger size, proteins, lipids, RNA and DNA molecules have been demonstrated [55-57].

line 21

function, whereas major scientific progress

line 30

anti-inflammatory cytokines [61].

  ?

line 32

a membrane lipid rearrangement indie occurs

lines 33-34

outer leaflet [62]. Thus, PS, a phospholipid normally localized

6 line 4

erythroblasts maturation [67].

line 7

further significance as a

line 8

tumoral marker [69].

Check more specially the references numbers in lines 9-14 !

lines 13-14

including microRNA and DNA [ref? ] to regulate……. ApoBDs in dead cells, MVs and exosomes contain fundamentally……..  primarily found in ApoBDs [ref ?].

line 26

a process closely involved in both cell clearance and intercellular communication [78, 79].

Author Response

1) We have modified the presentation of the paper, as suggested by the Reviewer 1 to make it easy to understand.

2)              We double-checked and expanded the references following the suggestions of the two reviewers.
